# Prediction of Postoperative Sepsis Based on Changes in Presepsin Levels of Critically Ill Patients with Acute Kidney Injury after Abdominal Surgery

**DOI:** 10.3390/diagnostics11122321

**Published:** 2021-12-09

**Authors:** Chang Hwan Kim, Eun Young Kim

**Affiliations:** Division of Trauma and Surgical Critical Care, Department of Surgery, Seoul St. Mary’s Hospital, College of Medicine, The Catholic University of Korea, Seoul 06591, Korea; changhwan.kim0227@gmail.com

**Keywords:** abdominal surgery, acute kidney injury, postoperative sepsis, presepsin

## Abstract

Presepsin (PSP) is a viable biomarker for the detection of bacterial infection, but it lacks accuracy when acute kidney injury (AKI) develops. Herein, we evaluated the diagnostic and prognostic value of PSP in predicting postoperative sepsis after abdominal surgery respective to the degree of AKI. A total of 311 patients who underwent abdominal surgery and were admitted to a surgical intensive care unit were enrolled and classified into non-AKI, mild-AKI (stage 1, stage 2 and stage 3 without renal replacement therapy (RRT)) and severe-AKI (stage 3 with RRT) group, according to the Kidney Disease Improving Global Outcomes criteria. In each group, PSP and other biomarkers were statistically analyzed between non-sepsis and postoperative sepsis at the admission (T0), 24 h (T1), 48 h (T2) and 72 h (T3) after surgery. In non-AKI and mild-AKI group, PSP levels were significantly higher in postoperative sepsis than non-sepsis group, whereas no difference was detected in the severe-AKI group. Cutoff values of PSP in the mild-AKI group for the prediction of postoperative sepsis were 544 pg/mL (AUC: 0.757, *p <* 0.001) at T0 and 458.5 pg/mL (AUC: 0.743, *p* < 0.001) at T1, significantly higher than in non-AKI group. In multivariate analysis, predictors of postoperative sepsis in the mild-AKI group were PSP at T2 (odds ratio (OR): 1.002, *p* = 0.044) and PSP at T3 (OR: 1.001, *p* = 0.049). PSP can be useful for predicting newly developed sepsis in patients with transient AKI after abdominal surgery with modified cutoff values.

## 1. Introduction

Sepsis from abdominal infection is a fatal condition which requires early diagnosis and appropriate treatment in order to avoid its high mortality and morbidity [1]. Initial resuscitation and source control of infection, including intervention or surgical procedure, are essential to improve survival outcome [1]. Surgical patients with abdominal infection carry an approximately 10% incidence of developing sepsis within 30 days after abdominal surgery and is associated with more than 20% mortality rate, despite the successful surgical elimination of source of infection [2]. The continued high risk of sepsis even after surgery for an abdominal infection could be partly explained by the patient’s vulnerability due to the physiological nature of postoperative condition. Hemodynamic instability due to fluid loss during surgery, and impaired cellular immune function induced by surgical stress, can lower the threshold of host’s barrier against postoperative infection [3]. Additionally, intraoperative manipulations of intestine or damaged tissue trigger non-infectious systemic inflammatory response syndrome (SIRS) with clinical manifestations similar to those of early stage sepsis [4]. It may interfere with and delay appropriate resuscitation via misdiagnosis or interfere with the detection of postoperative infectious complication [5]. Thus, an accurate and feasible diagnostic tool is needed to diagnose postoperative infection in a timely manner and also differentiate septic conditions from non-infectious SIRS.

Presepsin (PSP), a 13 kDa N-terminal fragment of soluble CD14 expressed in the acute phase of bacterial infection, represents a useful diagnostic biomarker in sepsis [6]. CD14 is a surface receptor expressed in macrophage or monocyte and produces complexes with bacterial ligand, such as lipopolysaccharide and lipoprotein binding protein [6]. Soluble CD14 subtype is then cleaved from the complex throughout the immune response, and we call this subtype PSP [7]. Since it is produced by direct interaction of bacteria ligands, such as lipopolysaccharide with CD14, a surface receptor on monocyte, it can be detected in early phase of bacterial infection [8]. Its diagnostic value is comparable to that of conventional markers, such as white blood cell (WBC), C-reactive protein (CRP) or procalcitonin (PCT), and can also be used to distinguish sepsis from non-infectious SIRS [1].

However, Nakamura et al. [9] reported that the diagnostic value of PSP can be unreliable when sepsis is accompanied by a severe form of acute kidney injury (AKI). The mechanism of this phenomenon is still unclear, but it is presumed to be related to the excretion of PSP through glomerulus, along with its molecular weight [10]. Considering that surgical patients with sepsis generally have faster progression of disease and higher mortality in the AKI state than in the non-AKI state, it is necessary to find an accurate diagnostic value of PSP classified according to the renal function. However, only few studies have evaluated the role of PSP level in postoperative patients according to the severity of AKI. Herein, we analyzed the change in PSP levels, depending on the severity of AKI based on the Kidney Disease Improving Global Outcomes (KDIGO) criteria in diagnosis of newly developed sepsis after abdominal surgery. Furthermore, we sought to determine the cutoff and predictive values of PSP in each grade of AKI and compare them with those of the conventional biomarkers.

## 2. Materials and Methods

### 2.1. Study Design and Patient Enrollment

Patients who underwent emergent surgery due to abdominal infection and were admitted to the surgical intensive care unit (SICU) of our institution were enrolled from July 2020 to March 2021. The scheme of patients’ selection and classification is provided in Figure 1. Inclusion criteria were patients who were (1) older than 18 years, (2) available with written consent and (3) treated via any type of abdominal surgery with successful removal of infection source. The definition of successful infection source control was the postoperative condition that included all of the following findings: (1) loss of clinical symptoms, including abdominal tenderness and rigidity; (2) recovery of bowel function; (3) body temperature below than 37.5 °C; and (4) WBC count less than 12,000 μL/L [11]. The exclusion criteria were (1) failure to remove the source of infection, (2) history of autoimmune diseases or immunodeficiency status or (3) “Do-Not-Resuscitate” consent. The failure to undergo infection-source removal was determined by clinical situations that included all of those findings: (1) ongoing or progressive abdominal tenderness and rigidity; (2) significant color change of surgical drain; (3) remaining infectious source confirmed by radiologic evaluation, such as abdominal computed tomography; and (4) worsening clinical course despite adequate treatment for sepsis. In the current study, the participants were divided into non-AKI and AKI groups, and the AKI group was subdivided into two groups according to the degree of severity based on KDIGO criteria: mild-AKI group, which includes stage 1, stage 2 and stage 3 without renal replacement therapy (RRT); and severe-AKI group, which includes stage 3 with RRT, as shown in (Figure 1) [12]. AKI was defined as occurring within 7 days after surgery or by discharge from SICU if discharge was within 7 days, and each grade of AKI was defined according to KDIGO criteria based on Kellum et al., as shown in Table 1 [12].

### 2.2. Data Recruitment and Postoperative Management

Patient demographics and clinical information were recorded prospectively. The perioperative outcome was also determined from the surgical record or medical charts, including type of surgery and the execution of dialysis or continuous renal replacement therapy. Inflammatory markers, including PSP, PCT and WBC, were sampled four times in each patient; first, at SICU admission after surgery (T0), and again 24 h after (T1), and 48 h after (T2), and 72 h after (T3) surgery. For each sample, 20 mL of whole blood was sampled in endotoxin-free tubes containing ethylenediaminetetraacetate, centrifuged for 10 min at 1600× *g* and then stored at −80 °C until analysis. PCT was measured by using an AFIAS PCT immunoassay (Boditech Med Inc., Chuncheon, Gangwon-do, Korea) [13]. PSP level was assessed by using a compact automated immunoanalyzer system (PATHFAST, Mitsubishi Chemical Europe GmbH, Düsseldorf, Germany) based on a chemiluminescent enzyme immunoassay according to the manufacturer’s instructions [14]. Serum creatinine and hourly urine output were collected until 7 days after surgery or the day of discharge from SICU if discharge was within 7 days. The scheme of data collection is demonstrated in Figure 2. Patients were regarded as eligible for discharge from SICU if they met all of the following criteria; (1) hemodynamic stability with mean arterial pressure higher than 65 mmHg without any vasopressor, (2) preserved respiratory function without ventilator, (3) cessation of continuous renal replacement therapy and conversion to dialysis, if needed, and (4) lactate level lower than 2 mmol/L.

### 2.3. Clinical Outcome Assessment 

According to the Third International Consensus Definitions for Sepsis and Septic Shock (Sepsis-3), sepsis was defined as life-threatening organ dysfunction caused by a dysregulated host response to infection [15]. Organ dysfunction was determined as a score of 2 or more on the Sequential Organ Failure Assessment (SOFA) score. Septic shock was defined as a clinical progression of sepsis with persistent hypotension requiring vasopressors to maintain mean arterial pressure higher than 65 mmHg and a serum lactate level higher than 2 mmol/L despite volume resuscitation [15]. In this study, we defined postoperative sepsis as both sepsis and septic shock newly developed after surgery with successful removal of infection source. As a result, patients without postoperative sepsis were denoted as group NS, and patients with postoperative sepsis were denoted as group PS. For the participants, sepsis was treated according to the Survival Sepsis Campaign bundle, via initial fluid resuscitation, appropriate application of vasopressors and empirical antibiotics, which were further adjusted to the results of bacterial culture [16]. Postoperative complications were defined as complications occurring within 7 days after surgery and were classified into grade III, IV and V based on the Clavien–Dindo classification that requiring surgical, endoscopic or radiologic intervention, or leading to life-threatening status or death [17]. We defined the ICU mortality as death during SICU stay, and the in-hospital mortality was defined as any death that occurred during the same hospitalization.

### 2.4. Statistical Analysis

Categorical variables were calculated by using the chi-square test or Fisher’s exact test. Continuous data are described as the median value with range or mean ± standard deviation, and the overall differences were assessed by using Student’s *t*-test or ANOVA test. The variables were tested to see whether they were normally distribution, using the Kolmogorov–Smirnov test, and in the case of variables not normally distributed, the Mann–Whitney test was used for a nonparametric test. The primary outcome was to analyze and establish a cutoff value of PSP, depending on the degree of renal impairment in the diagnosis of sepsis or septic shock developed after abdominal surgery. The secondary outcome was to compare the diagnostic value in predicting the postoperative sepsis of PSP with those of other biomarkers, such as PCT and WBC. Only significant variables in univariate analysis were used in multivariate regression analysis, expressed as the relative risk with the corresponding 95% confidence interval. To set the cutoff value of each inflammatory marker associated with postoperative sepsis, a receiver operating characteristic curve analysis was performed. The differences were regarded as statistically significant when *p* < 0.05. All statistical analyses were conducted by using SPSS statistical package software (version 24.0 for Windows; SPSS, Inc., Chicago, IL, USA).

## 3. Results

### 3.1. Demographics and Clinical Outcomes, and the Changes in Inflammatory Markers during Observational Period

During the study period, 368 patients were enrolled, and after excluding 57 patients based on our criteria, 311 patients were finally analyzed as Figure 1. Postoperative sepsis occurred in 139 patients (44.7%), including 99 patients with sepsis (31.8%) and 40 patients diagnosed with septic shock (12.9%). Demographic characteristics and clinical outcomes are summarized in Table 2. Mean age was 66.8 years (range 18–95), and 201 patients (67.5%) were male. The most common type of surgery was hepato-pancreato-biliary surgery (31.5%), followed by colorectal resection (17.7%). ICU mortality and in-hospital mortality occurred in 15 (4.8%) and 23 (7.4%) patients, respectively. Among all participants, 210 patients (67.5%) were classified as non-AKI, and postoperative sepsis occurred in 135 of them. Additionally, 101 patients (32.5%) were classified as AKI, including 64 patients manifesting postoperative sepsis. The number of patients in each AKI category is presented in Table 3 and Figure 1. The number of patients with postoperative sepsis in each AKI category was 28 of stage 1 (28/59, 47.5%), 8 cases of stage 2 (8/10, 80%), 18 cases of stage 3 without RRT (18/19, 94.7%) and 10 cases of stage 3 with RRT (10/13, 76.9%).

Median values of each inflammation marker at T0, T1, T2 and T3 are sequentially presented in Table 4 and in Figure 3. Table 5 describes the median values of each inflammatory marker under three groups depending on the severity of AKI; non-AKI, mild-AKI and severe-AKI. Severe-AKI group showed the highest median values of both PSP and PCT, followed by mild-AKI. The peak values of three groups were recorded at T3. However, the mean value of WBC did not differ between the two groups.

Based on Kellum et al. [12], the diagnosis of AKI was defined according to the KDIGO criteria: Stage 1 includes a 1.5-to-1.9-fold increase in serum creatinine compared to baseline, a more than 0.3 mg/dL increase in serum creatinine or urine output less than 0.5 mg/kg/h over 6 h. Stage 2 includes a 2.0-to-2.9-fold increase in serum creatinine or urine output less than 0.5 mg/kg/h over 12 h. Stage 3 includes a 3.0-fold increase in serum creatinine, a more than 4.0 mg/dL increase in serum creatinine, urine output less than 0.3 mg/kg/h over 24 h, anuria over 12 h or initiation of RRT. When creatinine criteria and urine output criteria showed different classification, a more severe value was adopted. Patients on dialysis with chronic kidney disease were classified as stage 3 with RRT.

### 3.2. Predictive Values of Various Markers in Postoperative Sepsis According to Renal Function

In Table 6 and Figure 4, the median values of each inflammatory marker were compared between group NS and group PS, according to AKI severity of participants. The median values of PSP in non-AKI and mild-AKI groups differed significantly between group NS and group PS during the observational period from T0, whereas the PCT level differed between group NS and group PS from T1. However, in the severe-AKI group, both PSP and PCT showed no difference during the observation period. On the contrary, the median value of WBC was similar between group NS and group PS during the entire study period, regardless of the patient’s renal function. 

The cutoff value for marker over time in the prediction of the development of postoperative sepsis according to AKI severity was determined by receiver operating characteristic curve analysis, and Table 7 summarizes the predictive value of each marker at different periods. In the non-AKI group, PSP had the highest predictive value, with a cutoff of 0.314 pg/mL at T0 (AUC: 0.686, *p* < 0.001) and a cutoff of 283 pg/mL at T1 (AUC: 0.644, *p* = 0.036). PCT had the highest predictive value of 0.045 ng/mL at T0 (AUC: 0.674, *p* < 0.001), followed by 0.075 ng/mL at T1 (AUC: 0.644, *p* = 0.001). In the mild-AKI group, PSP had the highest predictive value of 544 pg/mL T0 (AUC: 0.757, *p* < 0.001), followed by those of 458.5 pg/mL at T1 (AUC: 0.743, *p* < 0.001). PCT had the highest predictive value of 0.15 ng/mL at T0 (AUC: 0.861, *p* < 0.001), followed by 0.26 ng/mL at T1 (AUC: 0.784, *p* < 0.001). However, a significant cutoff value of both PSP and PCT in the severe-AKI group is unclear. Regarding WBC, it did not show a significant cutoff value in the non-AKI and mild-AKI groups.

PSP and PCT, which had significant cutoff values of postoperative sepsis, were analyzed to determine the predictive value of ICU mortality and in-hospital mortality, as shown in Table 8 and Table 9. Regarding ICU mortality, a PSP cutoff of 5052 pg/mL at T3 (AUC: 0.851, *p* = 0.042) and a PCT cutoff of 11.83 ng/mL at T3 (AUC: 0.825, *p* = 0.031) had the highest predictive value. In the case of in-hospital mortality, a PSP cutoff of 1437 pg/mL at T2 (AUC: 0.872, *p* < 0.001) and a PCT cutoff of 7.315 ng/mL at T2 (AUC: 0.857, *p* = 0.001) had the highest predictive value. The results of multiple regression analysis of factors associated with the risk of developing postoperative sepsis in each AKI severity group are presented in Table 10, Table 11 and Table 12, respectively. In the non-AKI group, the PSP level at T0 (odds ratio (OR): 1.001, *p* = 0.026), PSP level at T2 (OR: 1.002, *p* = 0.014), PCT level at T3 (OR: 1.824, *p* = 0.007) and SOFA score at T0 (OR: 1.292, *p* = 0.041) were the significant factors. In the mild-AKI group, PSP level at T2 (OR: 1.002, *p* = 0.044), the PSP level at T3 (OR: 1.001, *p* = 0.049) and APACHE-II score at T0 (OR: 1.297, *p* = 0.019) were significant factors for postoperative sepsis. In the severe-AKI group, however, no significant factor was identified in univariate analysis, and, therefore, no multivariate analysis was performed.

## 4. Discussion

In this study, the PSP levels in the non-AKI and mild-AKI groups were significantly higher in group PS than group NS throughout the observation period. In contrast, PCT did not show significant difference between the two groups at the time immediately after surgery (T0). Further, PSP levels at T0 and T2 and PCT levels at T3 were found to be the risk factors of postoperative sepsis in non-AKI group, and PSP levels at T2 and T3 were the risk factors in mild-AKI group.

Exposure of the abdominal cavity to extra-peritoneal space, massive fluid loss and tissue injury during abdominal surgery can decrease of peritoneal phagocyte function, resulting in local immuno-suppression inside the abdominal cavity [3]. Moreover, the surgical stress increases the secretion of cortisol, and the intestinal manipulation induces local inflammation of bowel wall or deterioration of the anatomical barrier of bowel; therefore, the bloodstream of the host turns out to be highly susceptible to bacterial translocation across the intestinal wall [3,18]. For these reasons, the onset of sepsis after abdominal surgery might be more rapid than in non-surgical conditions, and the hemodynamic instability triggered by intraoperative bleeding and fluid loss accelerates the progression to shock state [19]. Nevertheless, since the clinical features of early stage sepsis are similar to that of systemic inflammatory response after surgery, the early diagnosis of postoperative sepsis is difficult and the proper management can be often delayed until progression to a refractory state of shock [4,20,21]. Our results showed no significant difference in PCT levels at T0, the time immediately after surgery, between group NS and group PS under non-AKI and mild-AKI groups. PCT is released from C-cells of the parathyroid gland in normal circumstances, and pro-inflammatory cytokines, such as TNF-*α* and IL-6, increase its release from neuroendocrine cells in lungs or intestine [22,23]. Accordingly, PCT may be elevated not only by infection, but also by other sources of inflammation, such as trauma or a surgical procedure [5,9,24,25,26]. Because it can be induced by secondary response to pro-inflammatory cytokines, PCT has a limitation to be optimized early diagnostic marker of sepsis after abdominal surgery [22]. PSP, however, is the fragment of complex generated via direct interaction between a bacterial ligand, such as lipopolysaccharide, and the CD14 receptor of monocyte [6,26,27]. This immediate reaction enables the sensitive detection of bacterial infection and early diagnosis of sepsis [1,8,28,29]. Actually, the results of correlation suggest that, in contrast to PCT, the PSP level varied significantly in the postoperative sepsis group after surgery (T0) in non-AKI and mild-AKI groups. Thus, PSP can be used as an early diagnostic marker, compared with other markers, in predicting the development of sepsis after surgery for abdominal infection.

Further, we analyzed the diagnostic value of PSP in predicting postoperative sepsis in patients. In previous studies, PSP might be an inaccurate indicator for predicting sepsis in patients with renal dysfunction [9,10,30,31,32,33]. Although the precise mechanism has yet to be established, it is presumed that the clearance of PSP does not occur when renal function is decreased, since PSP is filtered in glomerulus or catabolized by proximal tubular cells [10]. Severe dehydration and tissue hypoxia due to tertiary space fluid loss can easily trigger AKI in patients with sepsis after abdominal surgery. Hence, a more practical application of PSP in the diagnosis of sepsis requires modified guidelines based on renal function stratification. Our results also revealed that the PSP level in severe-AKI group was very high (>2000 pg/mL) regardless of period or presence of postoperative sepsis. However, in the case of non-AKI and mild-AKI groups, a significance difference existed between group NS and group PS. The study of Nakamura et al. [31] demonstrated that PSP levels associated with stage 3 showed relatively lower accuracy in predicting sepsis when compared to stages 1 and 2, whereas our study found that PSP in stage 3 without RRT was effective in predicting postoperative sepsis. Additionally, we analyzed the cutoff values of PSP for predicting postoperative sepsis in each AKI group. The mild-AKI group showed a higher cutoff value of PSP at T0 and T1 than the non-AKI group. However, we failed to detect a significant level in the severe-AKI group. The markedly high values of PSP regardless of sepsis in severe-AKI group were associated with consistent hemodialysis that increases PSP by activating monocytes, the site of PSP synthesis and release [6,30,33]. As a result, PSP can be useful in predicting postoperative sepsis based on the modified cutoff value in non-AKI group, especially in the mild-AKI group. The fact that SICU patients undergoing emergent surgery often manifest a transient form of AKI suggests the clinical utility of PSP as a surveillance tool. A further study including a large number of samples should be conducted to corroborate our results in current study.

Additionally, our results revealed the risk factors of newly developed sepsis within 7 days after surgery. Since the early postoperative period could be the most vulnerable period for infection, the critical complications in this period can result in higher mortality. As mentioned above, the physiologic changes immediately after surgery complicate early diagnosis, suggesting the need for increased clinical attention regarding the patients with risk factors. We expect that our modified cutoff values of PSP by period would be used to tailor a sophisticated management strategy for sepsis after abdominal surgery. We further analyzed the cutoff values of PSP for ICU mortality and in-hospital mortality in the mild-AKI group. In both cases, PSP had significant cutoff values at T0 and T1, as well as a high PSP level of more than 1000 pg/mL for ICU mortality. Previous studies hypothesized that PSP can be used not only to predict the potential occurrence of sepsis but also mortality [1,34,35,36,37]. Similarly, our study results demonstrated that PSP may play a pivotal role in predicting ICU and in-hospital mortality, as well as the onset of sepsis during early postoperative phase, even in patients with mild AKI.

Despite these interesting findings, our study has inevitable limitations that need to be considered for proper interpretation. First, due to its non-randomized design, the risk of selection bias is inevitable, and the patients might not have received an equally standardized treatment protocol. However, a single clinician has been treating all SICU patients regardless of surgery type at our institution since 2016. Moreover, the treatment strategy remained unchanged during the study period [16]. Second, no subgroup analysis based on the type of abdominal organs or the difference in bacterial strains was performed. Furthermore, PSP values on the day of hemodialysis or non-hemodialysis may vary. To overcome these limitations, a prospective randomized controlled trial with a larger number of participants should be conducted in the near future, along with an additional subgroup analysis of each abdominal organ and hemodialysis profile. However, to our knowledge, the current study is the first to analyze the predictive value of PSP according to the severity of AKI in patients after abdominal surgery. Since there is no certain biomarkers to differentiate postoperative sepsis from SIRS, our findings suggest that PSP may be a promising and valid diagnostic tool for predicting sepsis in the early postoperative phase. Further, the modified cutoff values of PSP with regard to postoperative renal function could be effectively implicated to surgical critically ill patients who have a high potential for AKI in a real clinical setting. 

## 5. Conclusions

PSP would be a useful predictive marker for newly developing sepsis after abdominal surgery, as well as a potent diagnostic tool that is directly applicable to the patients with transient AKI who do not require RRT. By applying cutoff values of PSP adjusted for renal dysfunction, more accurate and tailored guidelines for the prediction of newly developed sepsis in surgical patients can be expected.

## Figures and Tables

**Figure 1 diagnostics-11-02321-f001:**
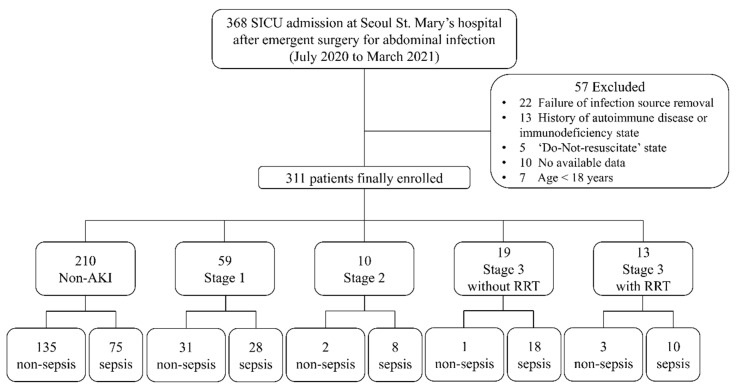
Schematic diagram of patient enrollment.

**Figure 2 diagnostics-11-02321-f002:**
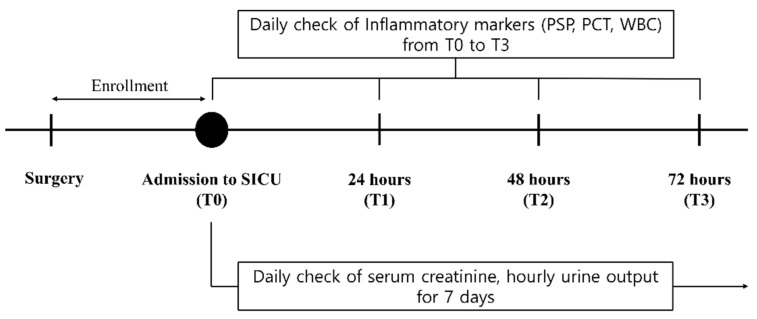
Schematic diagram of data collection.

**Figure 3 diagnostics-11-02321-f003:**
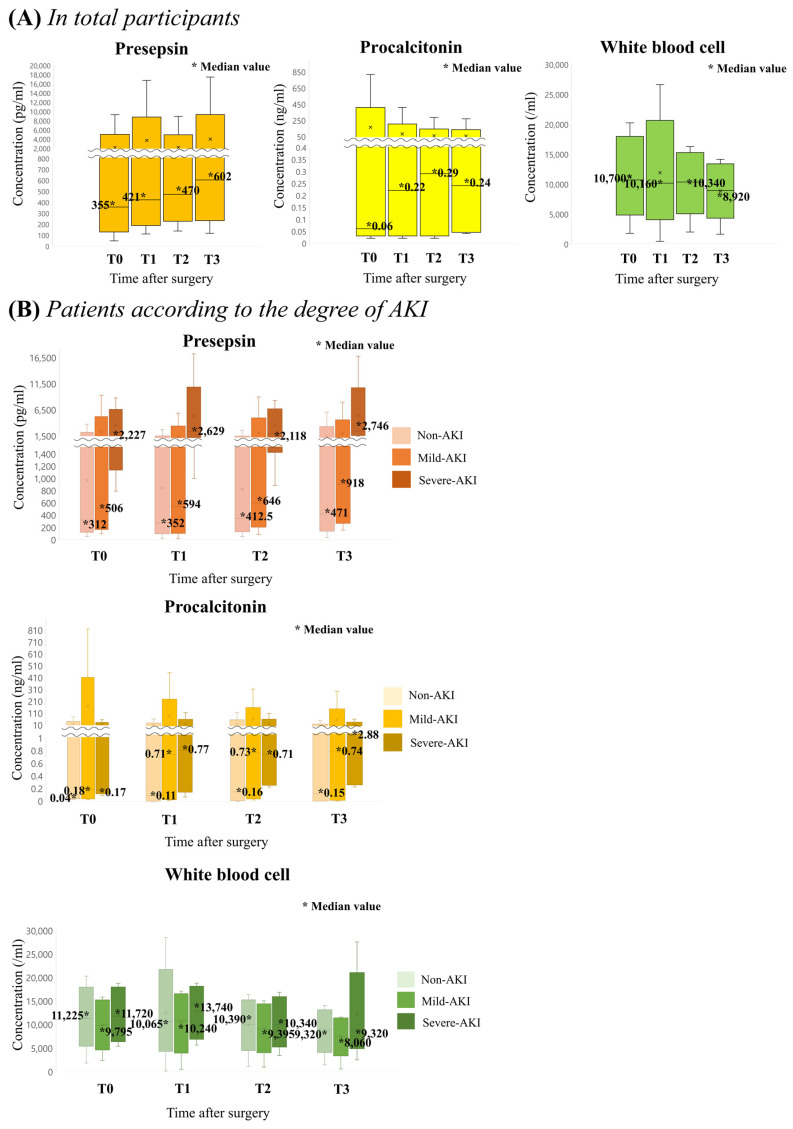
Comparison of median values of each inflammatory marker during observational period after surgery: (**A**) in total participants and (**B**) patients according to the degree of AKI.

**Figure 4 diagnostics-11-02321-f004:**
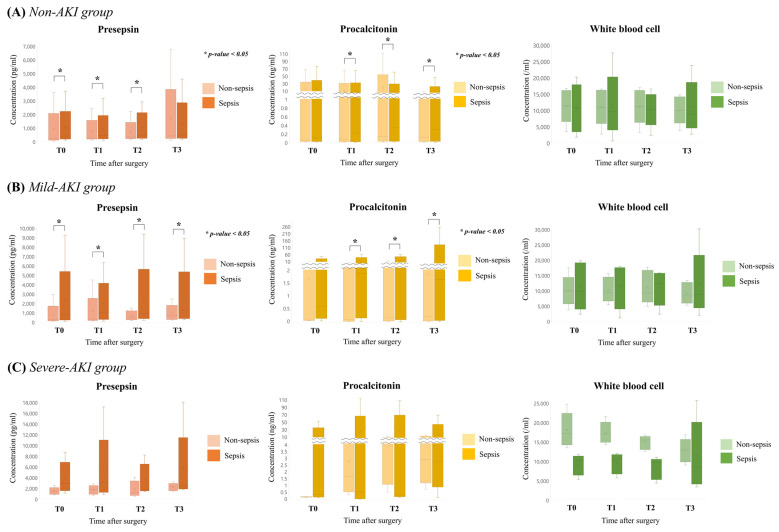
Comparison of median values of each inflammatory marker between groups with and without postoperative sepsis, according to AKI severity: (**A**) non-AKI group, (**B**) mild-AKI group and (**C**) severe-AKI group.

**Table 1 diagnostics-11-02321-t001:** Grading of AKI severity according to KDIGO criteria based on Kellum et al. [12].

Stage	Serum Creatinine	Urine Output
1	1.5-to-1.9-fold increase than baseline or more than 0.3 mg/dL increase	Less than 0.5 mL/kg/h over 6 h
2	2.0-to-2.9-fold increase than baseline	Less than 0.5 mL/kg/h over 12 h
3	3.0-fold increase than baseline or increase more than 4.0 mg/dL or initiation of renal replacement therapy	Less than 0.3 mL/kg/h over 24 h or anuria over 12 h

**Table 2 diagnostics-11-02321-t002:** Demographic characteristics and clinical outcomes of study population.

Variables	*N* = 311, Mean ± SD, (Range or %)
Demographic characteristics	
Age, years	66.8 ± 15.2 (18–95)
Gender (male/female)	210/101 (67.5/32.5)
BMI	23.3 ± 3.6 (13.4–38.7)
APACHE II score *	11.7 ± 6.3 (1–37)
SOFA score *	2.9 ± 2.8 (0–14)
Malignancy	143 (46)
Hypertension	158 (50.8)
Diabetes	99 (31.8)
Heart failure	7(2.3)
End-stage liver disease	9 (2.9)
Operative profiles	
Operative time, min	184.3 ± 132.5 (50–880)
Estimated blood loss, ml	323.6 ± 586.9 (50–2500)
Operation type	
Gastrectomy	20 (6.4)
Small bowel resection	27 (8.7)
Hepato-pancreatobiliary	98 (31.5)
Colorectal resection	55 (17.7)
Trauma	39 (12.5)
Miscellaneous	17 (5.5)
Clinical outcomes	
Sepsis	99 (31.8)
Septic shock	40 (12.9)
Length of ICU stay, day	2.6 ± 2.8 (1–16)
Length of hospital stay, day	15.2 ± 11.8 (1–93)
ICU mortality	15 (4.8)
In-hospital mortality	23 (7.4)

* This value was measured at the time of ICU admission. BMI, body mass index; APACHE II, acute physiology and chronic health evaluation; SOFA, sequential organ failure assessment; ICU, intensive care unit.

**Table 3 diagnostics-11-02321-t003:** AKI classification of patients after abdominal surgery.

Variables	*N* = 311 (Range or %)
Presence of Acute kidney injury *	
No-AKI	210 (67.5)
AKI	101 (32.5)
Grade of Acute kidney injury *	
Stage 1	59 (19)
Stage 2	10 (3.2)
Stage 3 without RRT	19 (6.1)
Stage 3 with RRT	13 (4.2)

AKI, acute kidney injury; ESKD, end-stage kidney disease. * Based on Kellum et al. [12], the diagnosis of AKI was defined according to the KDIGO criteria. Stage 1 includes a 1.5-to-1.9-fold increase in serum creatinine compared to baseline, a more than 0.3 mg/dL increase in serum creatinine or urine output less than 0.5 mg/kg/h over 6 h. Stage 2 includes a 2.0-to-2.9-fold increase in serum creatinine or urine output less than 0.5 mg/kg/h over 12 h. Stage 3 includes a 3.0-fold increase in serum creatinine, a more than 4.0 mg/dL increase in serum creatinine, urine output less than 0.3 mg/kg/h over 24 h, anuria over 12 h or initiation of RRT. When creatinine criteria and urine output criteria showed different classification, a more severe value was adopted. Patients on dialysis with chronic kidney disease were classified as stage 3 with RRT.

**Table 4 diagnostics-11-02321-t004:** Median values of inflammatory markers during observational period after surgery in total patients.

Inflammatory Markers	T0	T1	T2	T3
Presepsin	355 (46–9229)	421 (109–16,718)	470 (136–8875)	602 (116–17,428)
Procalcitonin	0.06 (0.02–816.4)	0.22 (0.02–407.4)	0.29 (0.02–285.1)	0.24 (0.04–268.6)
WBC	10,700 (1780–20,230)	10,160 (440–26,620)	10,340 (1990–16,260)	8920 (1630–14,150)

AKI, acute kidney injury; WBC, white blood cell; T0, immediately after surgery; T1, 24 h after surgery; T2, 48 h after surgery; T3, 72 h after surgery.

**Table 5 diagnostics-11-02321-t005:** Median values of inflammatory markers during observational period after surgery according to AKI severity: Non-AKI vs. mild-AKI (stage 1, stage 2 and stage 3 without RRT) vs. severe-AKI (stage 3 with RRT).

Inflammatory Markers	Renal Function, *n*	T0	T1	T2	T3
Presepsin	No-AKI, *n* = 210	312 (46–3690)	352 (109–3042)	412.5 (136–2790)	471 (116–6479)
	Mild-AKI, *n* = 88	506 (92–9229)	594 (109–5975)	646 (171–8875)	918 (230–8431)
	Severe-AKI, *n* = 13	2227 (785–8725)	2629 (1023–16,718)	2118 (914–8212)	2746 (1771–17,428)
Procalcitonin	No-AKI, *n* = 210	0.04 (0.02–75.55)	0.11 (0.02–63)	0.16 (0.02–112)	0.15 (0.04–49.37)
	Mild-AKI, *n* = 88	0.18 (0.02–816.37)	0.71 (0.04–407.43)	0.73 (0.04–285.05)	0.74 (0.04–268.57)
	Severe-AKI, *n* = 13	0.17 (0.08–52.35)	0.77 (0.09–110.26)	0.71 (0.23–104.35)	2.88 (0.24–61)
WBC	No-AKI, *n* = 210	11,225 (1780–20,230)	10,065 (440–26,620)	10,390 (2130–16,260)	9320 (2460–14,150)
	Mild-AKI, *n* = 88	9795 (2320–15,770)	10,240 (780–16,090)	9395 (1990–14,970)	8060 (1630–11,860)
	Severe-AKI, *n* = 13	12,720 (5350–11,720)	13,740 (5540–11,660)	10,340 (4270–10,640)	9320 (3420–26,730)

AKI, acute kidney injury; WBC, white blood cell; T0, immediately after surgery; T1, 24 h after surgery; T2, 48 h after surgery; T3, 72 h after surgery.

**Table 6 diagnostics-11-02321-t006:** Median values of inflammatory markers in patients with and without postoperative sepsis, according to AKI severity (non-AKI, *n* = 210; mild-AKI, *n* = 88; severe-AKI, *n* = 13).

Inflammatory Markers	Renal Function	T0, Median			T1, Median			T2, Median			T3, Median		
		No Sepsis or Septic Shock (*n* = 172)	Sepsis or Septic Shock (*n* = 139)	*p*-Value	No Sepsis or Septic Shock (*n* = 172)	Sepsis or Septic Shock (*n* = 139)	*p*-Value	No Sepsis or Septic Shock (*n* = 172)	Sepsis or Septic Shock (*n* = 139)	*p*-Value	No Sepsis or Septic Shock (*n* = 172)	Sepsis or Septic Shock (*n* = 139)	*p*-Value
Presepsin	Non-AKI	243	413.5	0.007	264	426	0.028	341	463.5	0.031	401	477	0.484
	Mild-AKI *	280	821	0.001	320.5	960	0.016	435	875.5	0.005	659	1226	0.008
	Severe-AKI *	1493	2895	0.300	2156	3291	0.370	1507	2182	0.486	2600.5	2746	0.424
Procalcitonin	Non-AKI	0.06	0.11	0.358	0.08	0.22	0.004	0.14	0.33	0.042	0.12	0.31	0.037
	Mild-AKI	0.04	0.56	0.124	0.09	1.79	0.012	0.15	1.96	0.005	0.19	1.63	0.018
	Severe-AKI	0.13	0.30	0.503	1.71	0.61	0.616	3.84	0.41	0.671	2.88	2.77	0.609
WBC	Non-AKI	11,430	10,850	0.666	10,370	9210	0.841	10,655	9835	0.450	9425	8430	0.055
	Mild-AKI	9935	9700	0.762	8965	11,190	0.912	8680	11,840	0.579	8055	8460	0.327
	Severe-AKI	17,120	10,155	0.608	16,440	10,195	0.608	14,950	9420	0.607	13,395	8840	0.764

AKI, acute kidney injury; WBC, white blood cell; T0, immediately after surgery; T1, 24 h after surgery; T2, 48 h after surgery; T3, 72 h after surgery. * Mild-AKI includes stage 1, stage 2 and stage 3 without RRT; severe-AKI includes stage 3 with RRT.

**Table 7 diagnostics-11-02321-t007:** Accuracy of inflammatory markers in predicting postoperative sepsis according to AKI severity, by period.

Inflammatory Markers	Renal Function	T0			T1			T2			T3		
		Cutoff	AUC	*p*-Value	Cutoff	AUC	*p*-Value	Cutoff	AUC	*p*-Value	Cutoff	AUC	*p*-Value
Presepsin	Non-AKI	314	0.686	<0.001	283	0.644	0.036	933.5	0.569	0.147	890.5	0.498	0.964
	Mild-AKI *	544	0.757	<0.001	458.5	0.743	<0.001	548.5	0.706	0.004	949.9	0.714	0.004
	Severe-AKI *	2739	0.741	0.229	3534	0.7	0.31	1624	0.762	0.21	1925	0.7	0.439
Procalcitonin	Non-AKI	0.045	0.674	<0.001	0.075	0.644	0.001	1.135	0.606	0.024	0.665	0.619	0.024
	Mild-AKI	0.15	0.861	<0.001	0.26	0.784	<0.001	0.965	0.77	<0.001	0.37	0.745	0.001
	Severe-AKI	0.155	0.87	0.064	7.775	0.4	0.612	17.76	0.333	0.405	16.61	0.444	0.796
WBC	Non-AKI	11,790	0.469	0.451	24,745	0.425	0.077	7610	0.446	0.255	19,225	0.405	0.051
	Mild-AKI	18,690	0.484	0.804	11,080	0.577	0.233	8805	0.555	0.429	9105	0.486	0.847
	Severe-AKI	25,760	0.067	0.028	21,720	0.133	0.063	20,610	0.133	0.063	22,100	0.286	0.38

AKI, acute kidney injury; WBC, white blood cell; T0, immediately after surgery; T1, 24 h after surgery; T2, 48 h after surgery; T3, 72 h after surgery; AUC, area under curve. * Mild-AKI includes stage 1, stage 2 and stage 3 without RRT; Severe-AKI includes stage 3 with RRT.

**Table 8 diagnostics-11-02321-t008:** Accuracy of inflammatory markers in predicting ICU mortality according to AKI severity, by period.

Inflammatory Markers	Renal Function	T0			T1			T2			T3		
		Cutoff	AUC	*p*-Value	Cutoff	AUC	*p*-Value	Cutoff	AUC	*p*-Value	Cutoff	AUC	*p*-Value
Presepsin	Mild-AKI	1643	0.832	<0.001	1238	0.806	0.001	1355	0.823	0.015	5052	0.851	0.042
Procalcitonin	Mild-AKI	1.035	0.729	0.02	0.995	0.746	0.023	0.985	0.782	0.044	11.83	0.825	0.031
WBC	Mild-AKI	7900	0.464	0.7	885	0.37	0.229	17,995	0.319	0.178	22,695	0.333	0.332

AKI, acute kidney injury; WBC, white blood cell; T0, immediately after surgery; T1, 24 h after surgery; T2, 48 h after surgery; T3, 72 h after surgery; AUC, area under curve.

**Table 9 diagnostics-11-02321-t009:** Accuracy of inflammatory markers in predicting in-hospital mortality according to AKI severity, by period.

Inflammatory Markers	Renal Function	T0			T1			T2			T3		
		Cutoff	AUC	*p*-Value	Cutoff	AUC	*p*-Value	Cutoff	AUC	*p*-Value	Cutoff	AUC	*p*-Value
Presepsin	Mild-AKI	713	0.791	<0.001	651	0.772	0.001	1437	0.872	<0.001	949	0.794	0.019
Procalcitonin	Mild-AKI	0.43	0.765	0.002	1.235	0.78	0.002	7.315	0.857	0.001	0.785	0.835	0.002
WBC	Mild-AKI	2325	0.394	0.187	10,820	0.462	0.66	11,840	0.596	0.331	9145	0.494	0.954

AKI, acute kidney injury; WBC, white blood cell; T0, immediately after surgery; T1, 24 h after surgery; T2, 48 h after surgery; T3, 72 h after surgery; AUC, area under curve.

**Table 10 diagnostics-11-02321-t010:** Risk factors associated with postoperative sepsis within 7 days after surgery in non-AKI group.

	Univariate Analysis		Multivariate Analysis	
	OR (95% CI)	*p*-Value	OR (95% CI)	*p*-Value
Presepsin at T0	1.001 (1.000–1.002)	0.006	1.001 (1.000–1.002)	0.026
Presepsin at T1	1.001 (1.000–1.002)	0.006	0.998 (0.997–1.000)	0.090
Presepsin at T2	1.001 (1.000–1.002)	0.037	1.002 (1.000–1.003)	0.014
Procalcitonin at T0	4.339 (1.260–14.944)	0.020	2.252 (0.956–5.307)	0.063
Procalcitonin at T1	1.117 (1.017–1.226)	0.020	0.782 (0.624–1.080)	0.133
Procalcitonin at T3	1.291 (1.025–1.626)	0.030	1.824 (1.182–2.814)	0.007
SOFA score at T0	1.778 (1.439–2.198)	<0.001	1.292 (1.011–1.653)	0.041
APACHE-II score at T0	1.108 (1.041–1.179)	0.001	1.135 (0.987–1.304)	0.075

AKI, acute kidney injury; T0, immediately after surgery; T1, 24 h after surgery; T2, 48 h after surgery; T3, 72 h after surgery; OR, odds ratio; CI, confidence interval.

**Table 11 diagnostics-11-02321-t011:** Risk factors associated with postoperative sepsis within 7 days after surgery in mild-AKI group (stage 1, stage 2 and stage 3 without RRT).

	Univariate Analysis		Multivariate Analysis	
	OR (95% CI)	*p*-Value	OR (95% CI)	*p*-Value
Presepsin at T0	1.001 (1.000–1.002)	0.005	1.000 (0.998–1.001)	0.671
Presepsin at T1	1.001 (1.000–1.001)	0.034	1.002 (0.999–1.005)	0.142
Presepsin at T2	1.002 (1.000–1.003)	0.007	1.002 (1.000–1.004)	0.044
Presepsin at T3	1.001 (1.000–1.002)	0.017	1.001 (1.000–1.003)	0.049
Procalcitonin at T1	1.086 (1.015–1.162)	0.017	0.967 (0.899–1.041)	0.377
Procalcitonin at T2	1.120 (1.006–1.246)	0.038	0.963 (0.915–1.013)	0.145
SOFA score at T0	1.806 (1.331–2.451)	<0.001	1.009 (0.665–1.530)	0.966
APACHE-II score at T0	1.179 (1.075–1.294)	<0.001	1.297 (1.043–1.614)	0.019

AKI, acute kidney injury; T0, immediately after surgery; T1, 24 h after surgery; T2, 48 h after surgery; T3, 72 h after surgery; OR, odds ratio; CI, confidence interval.

**Table 12 diagnostics-11-02321-t012:** Risk factors associated with postoperative sepsis within 7 days after surgery in severe-AKI group (stage 3 with RRT).

	Univariate Analysis		Multivariate Analysis	
	OR (95% CI)	*p*-Value	OR (95% CI)	*p*-Value
Presepsin at T0	1.001 (0.999–1.002)	0.276	-	*-*
Presepsin at T1	1.001 (0.999–1.002)	0.380	-	*-*
Presepsin at T2	1.000 (0.999–1.001)	0.462	-	*-*
Presepsin at T3	1.000 (0.999–1.002)	0.528	-	*-*
Procalcitonin at T0	2.284 (0.711–5.322)	0.357	-	*-*
Procalcitonin at T1	1.031 (0.886–1.200)	0.695	-	*-*
Procalcitonin at T2	1.017 (0.943–1.097)	0.664	-	*-*
Procalcitonin at T3	1.031 (0.926–1.147)	0.580	-	*-*
SOFA score at T0	1.288 (0.696–2.384)	0.420	-	*-*
APACHE-II score at T0	1.044 (0.888–1.227)	0.603	-	*-*

AKI, acute kidney injury; T0, immediately after surgery; T1, 24 h after surgery; T2, 48 h after surgery; T3, 72 h after surgery; OR, odds ratio; CI, confidence interval.

## Data Availability

The datasets used and analyzed in the current study are available from the corresponding author upon reasonable request.

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
