# Peer review of "Prediction of Postoperative Sepsis Based on Changes in Presepsin Levels of Critically Ill Patients with Acute Kidney Injury after Abdominal Surgery"

_diagnostics, 2021, doi:10.3390/diagnostics11122321_

Round 1

Reviewer 1 Report

This paper study the Prediction of postoperative sepsis based on changes in presepsin levels of critically ill patients with acute kidney in-jury after abdominal surgery and PSP can be useful for predicting newly developed sepsis in patients with transient AKI after abdominal surgery with modified cutoff values. It is a new and interesting topic, but there are a few weaknesses that should be addressed in this paper. Therefore, I suggest the authors resubmit it after a major revision. My suggestions are as follows:

1)  Authors should enrich the literature review by addressing more relevant papers. Only 30 references are too low for this journal. At least consider more recent references(2020 and 2021).

2) Your introduction is too short, please explain more about your method in this part.

3) Please add a flowchart of your method in the first part of your paper for more clarification. Although the flowchart is beneficial, It’s also important to outline the methodology behind your method.

4) I strongly suggest that the paper be proofread and reread meticulously again, particularly in regard to the spelling and grammatical mistakes.

5) Please modify table 1 and put all in one column( for example gender (and male and female)).

6) Figure 2 is not visible. Please explain more about it.

7) Your conclusion is too short.

Reviewer 2 Report

The authors discribed the prediction of postoperative sepsis based on changes in presepsin levels of critically ill patients with acute kidney injury after abdominal surgery.

The paper is well written and it is important for the scientific community.

Some considerations

1) The introduction describes little data about the compound under study. Please provide more information about PSP

2) The KDGO criteria would be better understood if they were in a table or in a figure

3) Figure 2 is difficult to understand the text. Please provide a figure in a better resolution

4) The table 5 is very complicated. There are a lot of information in a little space. Please provide a clear way to present the table. 

Reviewer 3 Report

The general aim of the study is well presented in the introduction and supported by an adequate design of the study.  The  presentation of the results in the Figures 2 and 3 is quite difficult to interpretate , due to a very low quality of the graphics. I would suggest to improve the quality of resolution and enlarge the panels, which are too small to be visualized correctly in the present form. Similarly, I would suggest a substantial editing the the  formatting of the Tables , where the words are ofter interrupted and wrapped, and need to be adapted to the size of the table spaces. 

Round 2

Reviewer 1 Report

The authors precisely applied all my comments. This version is acceptable.

Author Response

Thank you for your kind comment. 

This manuscript is a resubmission of an earlier submission. The following is a list of the peer review reports and author responses from that submission.